# Body Image Perception and Satisfaction of Junior High School Students: Analysis of Possible Determinants

**DOI:** 10.3390/children10061060

**Published:** 2023-06-14

**Authors:** Huizi Song, Yepeng Cai, Qian Cai, Wen Luo, Xiuping Jiao, Tianhua Jiang, Yun Sun, Yuexia Liao

**Affiliations:** 1School of Nursing, School of Public Health, Yangzhou University, Yangzhou 225009, China; ocuteh@163.com (H.S.);; 2Affiliated Hospital of Yangzhou University, Middle Hanjiang Road, Hanjiang District, Yangzhou 225012, China

**Keywords:** body image, body image perception, body image satisfaction, body shape aesthetics, junior high school student

## Abstract

Body image (BI) is a multidimensional construct that refers to one’s perceptions of and attitudes toward one’s own physical characteristics. Adolescence is a critical developmental stage in which concerns about BI increase. Therefore, the present cross-sectional study aimed to evaluate body image and aesthetic body shape standards in a sample of middle school students living in China. The researchers gathered demographic information, as well as height and weight data, for their study. They used a body silhouette to assess body image perception and body shape aesthetics and calculated two indexes: BIP, which measures the accuracy of self-perception and the estimation of bodily dimensions, and BIS, which indicates the difference between an individual’s perceived and ideal body images. A total of 1585 students in three grades at two middle schools were included in the study (759 = female, mean age = 13.67 ± 0.90; 839 = male, mean age = 13.70 ± 0.90). The results showed that the BIP bias rate of middle school students was 55.7%, and the BI dissatisfaction rate was 81.0%. Females tended to overestimate their body shape and desire to be thinner compared to males. Students with a higher BMI grading were more prone to underestimating their body shape and aspiring to be thinner. Furthermore, 8.6% of students chose underweight as the ideal body type for boys, while 22.6% chose underweight as the ideal body type for girls. In conclusion, there are significant gender differences in the aesthetic standards of body shape, and adolescents believe that for women, a thin body shape is beautiful.

## 1. Introduction

Body image (BI) is a psychological term related to self-image that predominantly refers to the visual representation of one’s own body shape and size, regardless of their actual body shape and size, and also includes subjective perceptions, feelings, and thoughts about that representation [1]. BI is thus a multidimensional construct that refers to one’s perceptions of and attitudes toward one’s own physical characteristics. BI involves multiple components: the perceptual component (BI perception (BIP)), the attitudinal component (BI satisfaction (BIS)), the cognition component, and the behavioral component [2]. BIP is defined as the accuracy with which someone perceives their appearance and can estimate their bodily dimensions [3,4]. BIS represents the discrepancy between an individual’s perceptual and ideal body shapes [5,6].

Distorted body image perception can lead to harmful behaviors and negative psychological effects, including an unhealthy diet, low self-esteem, sadness, and suicidal thoughts [7,8]. It can also have a negative impact on peer relationships and is often associated with bullying behavior [9].

Adolescence is a crucial stage in the development of an individual towards adulthood that is marked by significant physical, psychological, and behavioral changes [10,11]. During this period, adolescents become increasingly concerned about their body image due to external pressures from peers and family [6,12,13]. These concerns are further reinforced by mass media and social stereotypes about the ideal body shape [6]. However, the existing literature in China primarily focuses on investigating adult body image, with little attention paid to adolescent body image. In addition, most studies have analyzed body dissatisfaction in terms of the difference between how individuals perceive their bodies and how they want them to be [14]. To date, only a few and outdated studies have explored the relationship between the aesthetic standards of body shape and the two sexes [15,16]. Further research is required to investigate the current state of adolescent body image and aesthetic standards regarding body shape, as well as the potential factors that influence them.

This study is the first of its kind to evaluate the current status of BI perception and satisfaction among adolescents in the southeastern region of China. The study aims to explore the possible factors that influence these perceptions through the use of the Figure Rating Scale. The research questions that guide this study include: (1) What is the consistency between perceptive body type and BMI grading in adolescents? (2) What is the current status of BI perception and satisfaction among adolescents, and what factors may influence these perceptions? (3) What are the perceptions of adolescent boys and girls regarding the standard body shape, and are there any gender differences?

## 2. Materials and Methods

### 2.1. Design

This was a cross-sectional study based on self-reported data. It was conducted in accordance with the Declaration of Helsinki and approved by the Clinical Research Ethics Committee of the Yangzhou University School of Nursing School of Public Health ([2021] No. 103). The project was conducted by the Women and Children’s Teaching and Research Department of Yangzhou University, and participants were recruited through Yangzhou Education Centers. Principals of participating schools received an information sheet on the nature and purpose of the research, and subjects and their parents or legal guardians were informed of the anonymous and voluntary nature of participation. Informed consent forms were signed by all participants.

### 2.2. Participants

All adolescents attending the school could answer the questionnaires. The inclusion criteria were attending the school, completing all the parts of the questionnaires, having parental written consent, and agreeing to participate. Students who did not meet these criteria were excluded from the study. Only fully completed questionnaires were analyzed, resulting in a total of 1585 students from three grades at two middle schools. The recovery rate was 97.84%, with 759 females (mean age = 13.67 ± 0.90) and 839 males (mean age = 13.70 ± 0.90) included in the study. The measurement data from 35 students were excluded due to incomplete data.

### 2.3. Outcomes

#### 2.3.1. Measures

The surveyors in this study were trained uniformly, and the measuring instruments were calibrated uniformly as well. The heights and weights of the research subjects were measured in accordance with the research rules of the China National Student Constitution and Health Survey (CNSSCH [17]. Participants were asked to wear light clothing and stand barefoot during the measurement period. Body mass index (BMI) was calculated using the weight/height^2^ (kg/m^2^) formula to assess the weight status of each participant. The adolescent body types were classified into four categories: “underweight”, “normal weight”, “overweight”, and “obese” based on the WHO’s Body-Mass-Index-For-Age (z-score) [18].

In this study, the researchers utilized the Ma figural stimuli, which were originally developed by Collins (1991) and modified for a Chinese population, to assess the children’s body image [19]. The Ma figural stimuli consist of two groups, each containing seven images of boys and girls with varying body types. The images are arranged randomly and are coded from 1 to 7, with 1 representing the thinnest body type and 7 representing the most obese. Each image corresponds to a BMI Z-score ranging from −III to III, with −III to −II indicating wasting, −I, 0, and I indicating normal weight, II indicating overweight, and III indicating obese. The survey consisted of questions regarding self-perception of body type, ideal body type perception, and standard body type perceptions for boys and girls. Specifically, the participants were asked to identify which graph most resembled their current body type, which picture they preferred their body type to be like, and which pictures they believed represented the most ideal body types for boys and girls. The questionnaire was administered by trained investigators and completed by the respondents themselves.

#### 2.3.2. Body Image Assessment

BIP and BIS are two measures used to assess body image perception and satisfaction. BIP is calculated by subtracting the actual BMI-Z score from the self-perceived body type value, where a value of 0 indicates accurate perception, a value of ≥1 indicates an overestimate, and a value of ≤−1 indicates an underestimate [20]. On the other hand, BIS is calculated by subtracting the desired self-body type value from the perceived self-body type value, where a value of 0 indicates body image satisfaction, a value of ≥1 indicates a desire to become thin, and a value of ≤−1 indicates a fear of becoming obese [21].

### 2.4. Statistical Analysis

The data were entered twice into an Excel 2019 form and verified for accuracy, with any missing samples excluded. The data were then imported into SPSS 26.0 software for statistical analysis. The measurement data were tested for normality and expressed as means ± standard deviations. Differences between groups were analyzed using an independent samples *t*-test. Count data were expressed as frequencies or percentages, and the *χ*^2^ test was used to analyze the body image and body shape aesthetic standards of junior high school students with different characteristics. To examine the association of ordered variables such as age and BMI grading with dichotomous variables (the presence or absence of each category of body image), a trend-square test was used. Cohen’s consistency test was utilized to assess the consistency between self-perceived body type and actual BMI grading. The Kappa values for consistency coefficient ranged from 0 to 1, where 0 indicated no consistency, <0.4 indicated poor consistency, and 0.8 to 1 indicated high consistency. The test level was set at a = 0.05.

## 3. Results

### 3.1. Characteristics of Participants

A survey was conducted on 1585 junior high school students aged 12–15 years, out of which 830 (52.37%) were boys and 755 (47.63%) were girls. Additionally, 802 (50.60%) of the surveyed students were only children. The study found that 193 (12.18%) boys and 55 (3.47%) girls were classified as obese based on their BMI. The mean height, weight, and BMI values of male students were significantly higher than those of female students (*p* < 0.001). Detailed information is shown in Table 1.

### 3.2. Current Status and Analysis of BIP

The study revealed that 2.84% of the participants were underweight, 21.01% were overweight, and 15.65% were obese. Among junior high school students, 6.37% perceived themselves as underweight, 14.89% as overweight, and 3.34% as obese based on the Ma figural stimuli analysis. Cohen’s consistency test was conducted to compare actual BMI gradings and self-perceived body types, showing a low consistency coefficient *Kappa* of 0.275 (Table 2).

The survey found that 55.71% of respondents had varying degrees of BIP bias, with 33.44% underestimating and 22.27% overestimating. Boys had a significantly higher rate of underestimating their body shape than girls (43.0% vs. 22.9%, *χ*^2^ = 71.753, *p* < 0.001), while girls had a higher rate of overestimating their body shape (15.2% vs. 30.1%, *χ*^2^ = 50.605, *p* < 0.001). The study also found statistically significant differences in BIP among gender, age, and BMI grading, as shown in Table 3. Further analysis using the trend-square test revealed that as the children’s ages increased, the rate of underestimation decreased, while the rate of overestimation increased. The *χ*^2^*_trend_* values were 33.676 and 16.988, respectively, and for both, *p_trend_* < 0.001). Additionally, as the children’s BMI ratings increased, the rate of underestimation gradually increased, while the rate of overestimation gradually decreased (*χ*^2^*_trend_* were 79.463 and 112.919, respectively; both *p_trend_* < 0.001).

### 3.3. Current Status and Analysis of BIS

In a study conducted among 1585 junior high school students, it was found that 81.01% of them were dissatisfied with their body shape. Out of these students, 66.37% wanted to be thinner, while 14.64% wanted to be more obese. Interestingly, 53.28% of students with a normal BMI wanted to be thinner. The dissatisfaction rate was higher among female students compared to male students (83.4% vs. 78.8%, *χ*^2^ = 5.553, *p* = 0.018). Similarly, a higher percentage of female students wanted to be slim compared to male students (76.2% vs. 57.5%, *χ*^2^ = 61.871, *p* < 0.001). The study found that there were no significant differences in body satisfaction among students of different age groups (*p* = 0.747), but there were significant differences between different genders and different BMI gradings (*p* < 0.001), as shown in Table 4. Further analysis using a trend-square test revealed that as the BMI grade of children increased, there was a corresponding increase in the proportion of children who expected to be thin, while the proportion of those who expected to be obese decreased gradually (*χ*^2^*_trend_* were 255.283 and 137.104, respectively; both *p_trend_* < 0.001).

### 3.4. Current Status of Aesthetic Standards of Body Shape

The results of the survey indicate that over 90% of junior high school students preferred slim and normal body shapes as the standard for both the same and opposite sex in the Ma body shape chart. Among the students, 8.58% chose the standard male body shape to be underweight, while 22.59% chose the female body shape to be underweight. A significant difference was observed between the number of girls and boys who chose the “underweight” figure as the standard female body shape (26.23% vs. 19.28%, *χ*^2^ = 10.916, *p* < 0.001), while there was no statistically significant difference in the choice of the standard male body shape between genders (*p* = 0.438), as shown in Table 5.

## 4. Discussion

This survey showed that the consistency between self-perceptive body shapes and the actual BMI gradings of junior high school students in Yangzhou was low and tended to be underestimated. It is noteworthy that there were significant differences in body image between genders and BMI gradings. The percentage of overweight/obese adolescent in this study was 36.66%, which was generally consistent with the 36.61% overweight/obese rate of children surveyed in Anhui Province, China in 2019 [22], but much higher than the average of 19.0% of children aged 6–17 years in China in 2020 [23]. Therefore, the rate of overweight obesity among junior high school students in Yangzhou needs urgent attention.

The study conducted revealed that BMI grading plays a significant role in the body image of junior middle school students. These findings align with the results of Panagiotis Plotas’s survey, which revealed that the more overweight the middle school students were, the more likely they were to underestimate their body shape [24]. Our study found that half of the overweight/obese students perceived their body shape to be normal. This could be attributed to the societal normalization of obesity [25]. The high incidence of overweight and obesity has resulted in a cognitive bias towards body types in which individuals perceive overweight and obese body types as normal. Those who underestimate their body shape continue to engage in obesogenic behaviors, further increasing the risk of overweight or obesity [26]. This may be due to the fact that these children have lower perceptions of being overweight, which may result in less motivation to control their eating habits and take steps towards weight loss [27]. The cycle of the “normalization of obesity—underestimation of obesity—normalization of obesity” can be observed. To effectively control obesity, it is crucial for children to have an accurate understanding of their body type.

The study conducted found that gender plays a significant role in the body image of junior middle school students, which is consistent with the findings of other researchers. Boys tend to underestimate their body weight, while girls tend to overestimate it [28,29]. Additionally, girls are more likely to be dissatisfied with their bodies and desire to lose weight, while boys want to gain muscle [30,31]. This could be due to social pressures related to the current ideal body types, with girls striving for thinner body shapes and engaging in weight-reducing behaviors, while boys aim for a more muscular physique [32,33]. According to sociocultural models of body dissatisfaction, the unrealistic and mostly unattainable body ideals imposed by social pressures such as mass media, family, and peers can lead individuals to buy into those ideals and attempt to mold their bodies to fit them [34]. This psychological mechanism, known as the internalization of body shape ideals (IBSI), is argued to be a key factor in explaining the onset of body dissatisfaction [35]. According to research, adolescents who overestimate their weight tend to consume more fruits and vegetables and less fast food [26]. Additionally, these individuals may have more restrictive eating habits, such as skipping breakfast or dinner, which aligns with previous studies [28]. It is possible that these adolescents engage in activities aimed at losing weight, both healthy (such as maintaining a balanced diet and regular exercise) and unhealthy (such as extreme physical activity and restrictive eating habits) [29]. Previous research has also indicated that body image concerns may lead to emotional difficulties among adolescent females [36]. This may be attributed to the societal emphasis on physical appearance, which can negatively impact self-esteem [37]. The internalization of weight and body shape concerns can further exacerbate negative well-being [38].

Our study found a significant gender difference in the aesthetic standards of body shape, with the general perception of the female body shape being “thin is beautiful”. The media’s promotion of “bony beauty” has misled people to form the aesthetic of “thinness as beauty for women” [39,40]. As the Internet continues to grow, the impact of online media on body image ideals is affecting younger and younger individuals [41,42]. This is especially concerning for secondary school students who are at a crucial stage of physical and psychological development. It is during this time that their perception of body shape and aesthetic standards are being shaped. Therefore, it is imperative for researchers and relevant organizations to prioritize the investigation of body shape aesthetics in primary and secondary schools.

## 5. Conclusions

The study demonstrates that adolescents exhibit a cognitive bias towards their body shape and are generally dissatisfied with it. Notably, there were significant differences in body image perception among middle school students based on gender and BMI grading. Furthermore, the study found a significant gender difference in aesthetic standards of body shape, with the general perception of the female body shape being “thin is beautiful”. To address this issue, it is recommended that health education on body image be conducted in primary and middle schools with the involvement of families, schools, and society. This will help students develop a healthy body image and promote their overall physical and mental well-being.

## Figures and Tables

**Table 1 children-10-01060-t001:** General information and physical examination information of junior high school students.

Variables		Boys (*n =* 839)	Girls (*n* = 759)	*χ*^2^/*t*	*p*
Age (y)		13.70 ± 0.90	13.67 ± 0.90	0.744	0.457
Height (cm)		168.15 ± 9.03	161.50 ± 6.37	72.979	<0.001
Weight (kg)		63.44 ± 16.59	55.17 ± 11.95	87.383	<0.001
BMI (kg/m^2^)		22.23 ± 4.66	21.06 ± 3.89	45.952	<0.001
Only child	Yes	385	417	12.378	<0.001
	No	445	338		
Nationality	Han nationality	823	746	0.481	0.488
	Ethnic minority	7	9		
BMI grading	Overweight	25	20	86.327	<0.001
	Normal weight	431	528		
	Overweight	181	152		
	Obese	193	55		

**Table 2 children-10-01060-t002:** Consistency between self-perceived body shapes and BMI grades of junior high school students (*n* = 1585).

BMI Grading	Self-Perceived Body Shape				
	Underweight	Normal Weight	Overweight	Obese	Total
Underweight	19	26	/	/	45
Normal weight	82	854	23	/	959
Overweight	/	243	84	6	333
Obese	/	72	129	47	248
Total	101	1195	236	53	1585

*Kappa* = 0.275.

**Table 3 children-10-01060-t003:** Univariate analysis of BIP of junior high school students (*n* = 1585).

Variables		Body Image Perception			*χ* ^2^	*p*
		Under- Estimate	Accuracy	Over- Estimate		
Gender	Boys	357 (43.0)	347 (41.8)	126 (15.2)	89.520	<0.001
	Girls	173 (22.9)	355 (47.0)	227 (30.1)		
Age (y)	12	63 (40.6)	70 (45.2)	22 (14.2)	28.038	<0.001
	13	193 (37.9)	215 (42.2)	101 (19.8)		
	14	196 (32.5)	272 (45.1)	135 (22.4)		
	15	78 (24.5)	145 (45.6)	95 (29.9)		
BMI- grading	Underweight	2 (4.4)	11 (24.4)	32 (71.1)	167.834	<0.001
	Normal weight	258 (26.9)	434 (45.3)	267 (27.8)		
	Overweight	144 (43.2)	149 (44.7)	40 (12.0)		
	Obese	126 (50.8)	108 (43.5)	14 (5.6)		

**Table 4 children-10-01060-t004:** Univariate analysis of BIS of junior high school students (*n* = 1585).

Variables		Body Image Satisfaction			*χ* ^2^	*p*
		Expect to Become Thin	Satisfied	Expect to Become Obese		
Gender	Boys	477 (57.5)	176 (21.2)	177 (21.3)	78.553	<0.001
	Girls	575 (76.2)	125 (16.6)	55 (7.3)		
Age (y)	12	112 (72.3)	23 (14.8)	20 (12.9)	3.478	0.747
	13	337 (66.2)	98 (19.3)	74 (14.5)		
	14	400 (66.3)	115 (19.1)	88 (14.6)		
	15	203 (63.8)	65 (20.4)	50 (15.7)		
BMI- grading	Underweight	8 (17.8)	7 (15.6)	30 (66.7)	343.205	<0.001
	Normal weight	511 (53.3)	258 (26.9)	190 (19.8)		
	Overweight	301 (90.4)	23 (6.9)	9 (2.7)		
	Obese	232 (93.5)	13 (5.2)	3 (1.2)		

**Table 5 children-10-01060-t005:** Esthetic standards of body shape between the two sexes of junior high school students (*n* = 1585).

Variables	Standard Body Shape for Male			*χ* ^2^	*p*	Standard Body Shape for Female			*χ* ^2^	*p*
	Underweight	Normalweight	Overweight /Obese			Underweight	Normalweight	Overweight /Obese		
Boys	64	759	7	2.712	0.438	160	634	36	39.486	<0.001
Girls	72	673	10			198	556	1		
Total	136	1432	17			358	1190	37		

## Data Availability

The data presented in this study are not available in accordance with Regulation (EU) of the European Parliament and of the Council 2016/679 of 27 April 2016 regarding the protection of natural persons regarding the processing of personal data and the free circulation of these data (RGPD).

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
