# Peer review of "Body Image Perception and Satisfaction of Junior High School Students: Analysis of Possible Determinants"

_children, 2023, doi:10.3390/children10061060_

Round 1

Reviewer 1 Report

Interesting manuscript, they manage to adequately respond to their research objective, however, I think it is necessary for the approval of this manuscript to add the strengths and weaknesses of this research

none

Author Response

请参阅附件

Reviewer 2 Report

Many thanks for authors for drafting the manuscript about this important topic. The research area is an important one and the results have a potential to add in understanding of youngsters’ body perception. However, the manuscript requires - in my opinion - considerable amount of work to improve the language expression. At the moment I feel that it was not possible to offer comprehensive evaluation due to language issues. In many places the language was very difficult to follow and in other places not neutral enough – but rather judgmental. Further comments can be found below.

Title

The usage of word „size” is somewhat problematic in here. Authors should consider rather using body image perception and satisfaction as in the abstract.

Abstract

While it is possible to get the gist of the research aims and results from the abstracts, the language is difficult to follow. Also, the authors should add the age group of the pupils – not only the years groups as these may differ between the countries. Small formal point is the use of percentages – it would be OK to report percentages as xx.x – not xx.xx. Finally, while undoubtedly the targeting education is important, it is a bit difficult to follow the conclusion – especially the mental health part. Was this related to cognitive deviation (and what is cognitive deviation in this instance) mentioned previously?

Introduction

There appears to be relevant information included in the introduction with authors specifying the research questions. However, and with my apologies, I find evaluation of the introduction and its quality very difficult due to language use and selected terminology. Just as an example what is cognitive body type? Does this mean self-perception? Further, language as standard body type is also difficult to understand. First paragraph was especially hard to follow.

Methods

Please could the authors add information about the age group of the participants – this is only included in the results. Short summary of inclusion and exclusion criteria and recruitment methods are recommendable in here. Also, a self-developed questionnaire is mentioned – but details are unclear about what this consisted of. Please could authors also elaborate about how the consent was obtained from the children (parental consent?). It also appears that the research was conducted as a part of a national survey – was participation compulsory?

It would be also good if the authors would add a sentence or two to better link the listed statistical methods to the precise research questions.

Results

Some language such as “fat” in should not be used in the text. Tables would need to be formatted so that no word is cut in the middle. Please could the authors check that the repetition between the tables and text is not too frequent.

Discussion

A number of new references have been introduced in the discussion. Please could the authors include these in the introduction also. Languagewise, the discussion is difficult to follow in a number of the places, making evaluation limited.

At the moment I feel that it was not possible to offer comprehensive evaluation due to language issues. In many places the language was very difficult to follow and in other places not neutral enough – but rather judgmental.

Author Response

请参阅附件

Round 2

Reviewer 2 Report

Dear Authors

I would like to thank you for the time that you have taken to engage with the feedback. In my view the manuscript and the language expression has considerably improved. While most the comments were satisfactorily addressed, I would like to highlight the following issues that would need some further attention.

Firstly, please ensure that all the abbreviations are explained at the first time (abstract BIP).

Secondly – the previous point about including considerable amount of new (relevant) literature in the discussion without including this in the discussion. This should be rectified – as the references discussed in the discussion are crucial for the readers to understand the context and purpose of the manuscript.

Thirdly, the manuscript would benefit from a language editing to ensure correct use of terminology and language. For example – in the tables word satisfaction is used – better would be e.g. satisfied.

Fourthly, considering the emphasis on mental and physical well-being in the conclusion, it would have been good to connect this shortly in discussion, based on the literature in introduction.

I would recommend one more through english check.

Author Response

请参阅附件。
